# LEARNING DE-BIASED REPRESENTATIONS WITH BIASED REPRESENTATIONS

## ABSTRACT

Many machine learning algorithms are trained and evaluated by splitting data from a single source into training and test sets. While such focus on *in-distribution* learning scenarios has led interesting advances, it has not been able to tell if models are relying on dataset biases as shortcuts for successful prediction (e.g., using snow cues for recognising snowmobiles). Such biased models fail to generalise when the bias shifts to a different class. The *cross-bias generalisation* problem has been addressed by de-biasing training data through augmentation or re-sampling, which are often prohibitive due to the data collection cost (e.g., collecting images of a snowmobile on a desert) and the difficulty of quantifying or expressing biases in the first place. In this work, we propose a novel framework to train a de-biased representation by encouraging it to be *different* from a set of representations that are biased by design. This tactic is feasible in many scenarios where it is much easier to define a set of biased representations than to define and quantify bias. Our experiments and analyses show that our method discourages models from taking bias shortcuts, resulting in improved performances on de-biased test data.

## 1 INTRODUCTION

Most machine learning algorithms are trained and evaluated by randomly splitting a single source of data into training and test sets. Although this is a standard protocol, it is blind to a critical problem: the existence of dataset bias (Torralba & Efros, 2011). For instance, many frog images are taken in swamp scenes, but swamp itself is not a frog. Nonetheless, a neural network will exploit this bias (i.e., take "shortcuts") if it yields correct predictions for the majority of training examples. If bias is sufficient to achieve high accuracy, there is little motivation for models to learn the complexity of the intended task, despite its full capacity to do so. Consequently, a model that relies on bias will achieve high in-distribution accuracy, yet fail to generalise when the bias shifts.

We tackle this "cross-bias generalisation" problem where a model does not exploit its full capacity due to the "sufficiency" of bias cues for prediction of the target label in training data. For example, language models make predictions based on the presence of certain words (e.g., "not" for "contradiction") (Gururangan et al., 2018) without much reasoning on the actual meaning of sentences, even if they are in principle capable of sophisticated reasoning. Similarly, convolutional neural networks (CNNs) achieve high accuracies on image classification by using local texture cues as shortcut, as opposed to more reliable global shape cues (Geirhos et al., 2019; Brendel & Bethge, 2019).

Existing methods attempt to remove a model's dependency on bias by de-biasing the training data through data augmentation (Geirhos et al., 2019) or re-sampling tactics (Li & Vasconcelos, 2019). Others have introduced a pre-defined set of biases that a model is trained to be independent against (Wang et al., 2019). These prior works assume that bias can easily be defined or quantified, but real-world biases often do not (e.g., texture bias above).

To address this limitation, we propose a novel framework to train a de-biased representation by encouraging it to be "different" from a set of representations that are biased by design. Our insight is that biased representations can easily be obtained by utilising models of smaller capacity (e.g., bag of words for word bias and CNNs of small receptive fields for texture bias). Experiments show that our method is effective in reducing a model's dependency on "shortcuts" in training data, as evidenced by improved accuracies in test data where the bias is either shifted or removed.

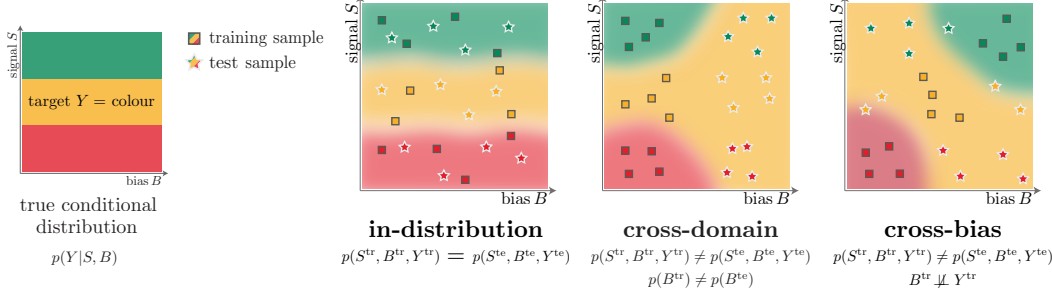

**Figure 1: Learning scenarios**. Different distributional gaps may take place between training and test distributions. Our work is addressing the *cross-bias generalisation* problem. Background colours on the right three figures indicate the decision boundaries of models trained on given training data.

## 2 PROBLEM DEFINITION

We provide a rigorous definition of our over-arching goal: overcoming the bias in models trained on biased data. We show that the problem we tackle is novel and realistic.

### 2.1 CROSS-BIAS GENERALISATION

We first define random variables, signal $S$ and bias $B$ as cues for the recognition of an input $X$ as certain target variable $Y$. Signals $S$ are the cues essential for the recognition of $X$ as $Y$; examples include the shape and skin patterns of frogs for frog image classification. Biases $B$'s, on the other hand, are cues not essential for the recognition but correlated with the target $Y$; many frog images are taken in swamp scenes, so swamp scenes can be considered as $B$. A key property of $B$ is that intervening on $B$ should not change $Y$; moving a frog from swamp to a dessert scene does not change the "frogness". We assume that the true predictive distribution $p(Y|X)$ factorises as $\int p(Y|S,B)p(S,B|X)$, signifying the sufficiency of $p(S,B|X)$ for recognition.

Under this framework, three learning scenarios are identified depending on the change of relationship $p(S,B,Y)$ across training and test distributions, $p(S^{\text{tr}}, B^{\text{tr}}, Y^{\text{tr}})$ and $p(S^{\text{te}}, B^{\text{te}}, Y^{\text{te}})$, respectively: in-distribution, cross-domain, and cross-bias generalisation. See Figure 1 for a summary.

**In-distribution.** $p(S^{\text{tr}}, B^{\text{tr}}, Y^{\text{tr}}) = p(S^{\text{te}}, B^{\text{te}}, Y^{\text{te}})$. This is the standard learning setup utilised in many benchmarks by splitting data from a single source into training and test data at random.

**Cross-domain.** $p(S^{\text{tr}}, B^{\text{tr}}, Y^{\text{tr}}) \neq p(S^{\text{te}}, B^{\text{te}}, Y^{\text{te}})$ and furthermore $p(B^{\text{tr}}) \neq p(B^{\text{te}})$. $B$ in this case is often referred to as "domain". For example, training data consist of images with ($Y^{\text{tr}}$=frog, $B^{\text{tr}}$=wilderness) and ($Y^{\text{tr}}$=bird, $B^{\text{tr}}$=wilderness), while test data contain ($Y^{\text{te}}$=frog, $B^{\text{te}}$=indoors) and ($Y^{\text{te}}$=bird, $B^{\text{te}}$=indoors). This scenario is typically simulated by training and testing on different datasets (Ben-David et al., 2007).

**Cross-bias.** $p(B^{\text{tr}}) \not\perp\!\!\!\perp p(Y^{\text{tr}})$[1] and the dependency changes across training and test distributions: $p(B^{\text{tr}}, Y^{\text{tr}}) \neq p(B^{\text{te}}, Y^{\text{te}})$. We further assume that $p(B^{\text{tr}}) = p(B^{\text{te}})$, to clearly distinguish the scenario from the cross-domain generalisation. For example, training data only contain images of two types ($Y^{\text{tr}}$=frog, $B^{\text{tr}}$=swamp) and ($Y^{\text{tr}}$=bird, $B^{\text{tr}}$=sky), but test data contain unusual class-bias combinations ($Y^{\text{te}}$=frog, $B^{\text{te}}$=sky) and ($Y^{\text{te}}$=bird, $B^{\text{te}}$=swamp). Our work addresses this scenario.

### 2.2 EXISTING CROSS-BIAS GENERALISATION METHODS AND THEIR ASSUMPTIONS

Under cross-bias generalisation scenarios, the dependency $p(B^{\text{tr}}) \not\perp\!\!\!\perp p(Y^{\text{tr}})$ makes bias $B$ a viable cue for recognition. The model trained on such data becomes susceptible to interventions on $B$, limiting its generalisabililty when the bias is changed or removed in the test data. There exist prior approaches to this problem, but with different types and amounts of assumptions on $B$. We briefly recap the approaches based on the assumptions they require. In the next part §2.3, we will define our novel problem setting that requires an assumption distinct from the ones in prior approaches.

---

[1] $\perp\!\!\!\perp$ and $\not\perp\!\!\!\perp$ denote independence and dependence, respectively.

**When an algorithm to disentangle bias $B$ and signal $S$ exists.** Being able to disentangle $B$ and $S$ lets one collapse the feature space corresponding to $B$ in both training and test data. A model trained on such normalised data then becomes free of biases. As ideal as it is, building a model to perfectly disentangle $B$ and $S$ is often unrealistic.

**When a generative algorithm or data collection procedure for $p(X|B)$ exists.** When additional examples can be supplied through $p(X|B)$, the training dataset itself can be de-biased, i.e., $B \perp\!\!\!\perp Y$. For example, one can either collect or synthesise unusual images like frogs in sky and birds in swamp to balance out the bias. Such a data augmentation strategy is indeed a valid solution adopted by many prior studies (Panda et al., 2018; Geirhos et al., 2019; Shetty et al., 2019). However, collecting unusual inputs can be expensive (Peyre et al., 2017), and building a generative model with pre-defined bias types (Geirhos et al., 2019) may suffer from bias mis-specification and the lack of realism.

**When a predictive algorithm or ground truth for $p(B|X)$ exists.** Conversely, when one can tell the bias $B$ for every input $X$, two approaches are feasible. (1) The first is a data re-weighting solution: we give greater weights on frogs in sky than frogs in swamps to even out the correlation in $p(B, Y)$ (Li et al., 2018; Li & Vasconcelos, 2019). (2) The second approach removes the dependency between the model predictions $f(X)$ and the bias $B$. Many existing approaches for fairness in machine learning have proposed independence-based regularisers to encourage $f(X) \perp\!\!\!\perp B$ (Zemel et al., 2013) or the conditional independence $f(X) \perp\!\!\!\perp B \,|\, Y$ (called the "separation" constraint, Hardt et al. (2016)). Other approaches have proposed to remove predictability of $p(B|X)$ based on $f(X)$ through domain adversarial losses (Li & Vasconcelos, 2019; Wang et al., 2019) or projection (Wang et al., 2019; Quadrianto et al., 2019).

The knowledge on $p(B|X)$ is provided in many realistic scenarios. For example, when the aim is to remove gender biases $B$ in a job application process $p(Y|X)$, applicants' genders $p(B|X)$ are supplied as ground truths. However, there exist cases when $B$ is difficult to even be defined or quantified but can only be indirectly specified. We tackle such a scenario in the next part.

### 2.3 OUR SCENARIO: CAPTURING BIAS WITH A PARTICULAR SET OF MODELS

Under the cross-bias generalisation scenario, certain types of biases are not easily addressed by the above methods. Take texture bias as an example (§1, Geirhos et al. (2019)): (1) texture $B$ and shape $S$ cannot easily be disentangled, (2) building a generative model $p(X|B)$ or collecting unusual images is expensive, (3) building the predictive model $p(B|X)$ for texture requires enumeration (classifier) or embedding (regression) of all possible textures, which is not feasible.

However, slightly modifying the third assumption results in a problem setting that allows interesting application scenarios. Instead of assuming explicit knowledge on $p(B|X)$, we approximate $B$ by defining a set of models $G$ that are biased towards $B$ by design. For texture biases, for example, we define $G$ to be the set of convolutional neural network (CNN) architectures with $\leq 5 \times 5$ overall receptive fields. Then, any learned model $g \in G$ can by design make predictions $g(x)$ based on the patterns that can only be captured with small receptive fields (i.e., textures).

More precisely, we define $G$ to be a **bias-characterising model class** for the bias-signal pair $(B, S)$ if for every possible joint distribution $p(B, X)$ there exists a $g \in G$ such that $p(B|X) \approx g(X)$ (**recall condition**) and every $g \in G$ satisfies $g(X) \perp\!\!\!\perp S \,|\, B$ (**precision condition**). In practice, $G$ may not necessarily include all biases and may also capture important signals (i.e., imperfect recall and precision). With this in mind, our framework is formulated so that $f(X)$ does not ignore signals captured by $G$ – we do not require G to be perfect. Further detail is provided in §2.3.

There exist many scenarios when such $G$ can be defined, as we are given several evidence for the type of bias. For instance, action recognition models have been reported to rely heavily on static cues without learning temporal cues (Li et al., 2018; Li & Vasconcelos, 2019); though the actual bias may not precisely be *any static cue*, we can still regularise the 3D convolutional networks towards better generalisation across static cue biases by defining G to be the set of 2D convolutional architectures. It has been argued that visual question answering (VQA) models, too, rely overly on language biases rather than the visual cues (e.g. without looking at the image, one knows the answer to what colour is the banana is yellow) (Agrawal et al., 2018). We can define G as the set of models looking at

the language modality only. Entailment models are biased towards the presence of certain words (e.g. when there are many nots, the sentence is contradictory), rather than really understanding the underlying meaning of sentences (McCoy et al., 2019; Niven & Kao, 2019). We can design G to be the set of bag-of-words classifiers (He et al., 2019; Clark et al., 2019). Generally, these scenarios exemplify situations when the added architectural capacity is not fully utilised due to the sufficiency of simpler cues for solving the task in the given training set.

## 3 PROPOSED METHOD

We present a solution for the cross-bias generalisation when the bias-characterising model class $G$ is known (see §2.3); the method is referred to as REBI'S[2]. The solution consists of training a model $f$ for the task $p(Y|X)$ with a regularisation term encouraging the independence between the prediction $f(X)$ and the set of all possible biased predictions $\{g(X) \,|\, g \in G\}$. We will introduce the precise definition of the regularisation term and discuss why and how it leads to the unbiased model.

### 3.1 REBI'S: REMOVING BIAS WITH BIAS

If $p(B|X)$ is fully known, we can directly encourage $f(X) \perp\!\!\!\perp B$. Since we only have access to the set of biased models $G$ (§2.3), we seek to promote $f(X) \perp\!\!\!\perp g(X)$ for every $g \in G$. Simply put, we de-bias a representation $f \in F$ by designing a set of biased models $G$ and letting $f$ run away from $G$. This leads to the independence from bias cues $B$ while leaving signal cues $S$ as valid recognition cues; see §2.3. We will specify REBI'S learning objective after introducing our independence criterion, HSIC.

**Hilbert-Schmidt Independence Criterion (HSIC).** Since we need to measure degree of independence between continuous random variables $f(X)$ and $g(X)$ in high dimensional spaces, it is infeasible to resort to histogram-based measures; we use HSIC (Gretton et al., 2005). For two random variables $U$ and $V$ and kernels $k$ and $l$, HSIC is defined as $\text{HSIC}^{k,l}(U,V) := ||C_{UV}^{k,l}||_{\text{HS}}^2$ where $C^{k,l}$ is the cross-covariance operator in the Reproducing Kernel Hilbert Spaces (RKHS) of $k$ and $l$ (Gretton et al., 2005), an RKHS analogue of covariance matrices. $|| \cdot ||_{\text{HS}}$ is the Hilbert-Schmidt norm, a Hilbert-space analogue of the Frobenius norm. It is known that for two random variables $U$ and $V$ and radial basis function (RBF) kernels $k$ and $l$, $\text{HSIC}^{k,l}(U,V) = 0$ if and only if $U \perp\!\!\!\perp V$. A finite-sample estimate $\text{HSIC}_0^{k,l}(U,V)$ has been used in practice for statistical testing (Gretton et al., 2005; 2008), feature similarity measurement (Kornblith et al., 2019), and model regularisation (Quadrianto et al., 2019; Zhang et al., 2018). $\text{HSIC}_0^{k,l}(U,V)$ with $m$ samples is defined as $(m-1)^{-2}\text{tr}(\widetilde{U}\widetilde{V}^T)$ where $\widetilde{U}$ is a mean-subtracted matrix of pairwise kernel similarities $\widetilde{U}_{ij} = k(u_i, u_j) - m^{-1}\sum_{j'} k(u_i, u_{j'})$ among samples $\{u_i\} \sim U$. $\widetilde{V}$ is defined similarly.

**Minimax optimisation for bias removal.** In our case, we compute

$$\text{HSIC}_0^k(f(X), G(X)) := \max_{g \in G} \text{HSIC}_0^k(f(X), g(X)) \qquad (1)$$

with an RBF kernel $k$ for the degree of independence between representation $f \in F$ and the biased representations $G$. We write $\text{HSIC}_0(f, G)$ and $\text{HSIC}_0(f, g)$ as shorthands. Since the problem $\min_f \text{HSIC}_0(f, G)$ allows trivial solutions $f = \text{const}$, we use the **canonical kernel alignment (CKA)** (Shawe-Taylor & Cristianini, 2004; Kornblith et al., 2019) criterion defined by $\text{CKA}_0(f, g) = \text{HSIC}_0(f, g)/(\text{HSIC}_0(f, f)^{\frac{1}{2}} \text{HSIC}_0(g, g)^{\frac{1}{2}})$. The learning objective for $f$ is then defined as

$$\min_{f \in F} \left\{ \mathcal{L}(f, X, Y) + \lambda \max_{g \in G} \text{CKA}_0(f, g) \right\}, \qquad (2)$$

where $\mathcal{L}(f, X, Y)$ is the loss for the main task $p(Y|X)$ and $\lambda > 0$. We consider replacing the inner optimisation with $L_2$ minimisation $\min_{g \in G} ||f - g||_2$, while retaining the CKA regularisation for the outer optimisation for $f$. Intuitively, $L_2$ minimisation poses a stronger similarity condition between $f$ and $g$ (in fact identity) than does $CKA$, leading to better de-biasing performances (§4.2.3).

---

[2] Abbreviation of "removing bias"; pronounced like Levi's.

**Independence versus separation.** The CKA regularisation in equation 2 encourages $f(X) \perp\!\!\!\perp g(X)$. This may lead to less stable optimisation as removing bias often increases the main task loss $\mathcal{L}$. Furthermore, $g(X)$ may also capture signals along with bias (imperfect precision), and suppressing important signals is not desirable. To avoid such cases, we formulate our objective as conditional independence $f(X) \perp\!\!\!\perp g(X) | Y$ by computing the **separation CKA**, $\mathrm{SCKA}_0(f, g) = |\mathcal{Y}|^{-1} \sum_{y \in \mathcal{Y}} \mathrm{CKA}_0(f(X|Y = y), g(X|Y = y))$, where the term "separation" (or "equalised odds") comes from the fairness literature (Hardt et al., 2016). Unlike independence that requires $f(X)$ to ignore $g(X)$ altogether, conditional independence allows $f(X)$ to utilise cues captured by $g(X)$ (i.e., whether it is bias or signal) if it highly correlates with Y. In other words, `REBI'S` encourages $f(X)$ to learn features beyond those already captured by $g(X)$, so that it can generalise well to test sets with different biases (i.e., cross-bias generalisation).

The final learning objective for `REBI'S` is then

$$\min_{f \in F} \left\{ \mathcal{L}(f, X, Y) + \lambda \max_{g \in G} \mathrm{SCKA}_0(f, g) \right\}. \tag{3}$$

## 3.2 WHY AND HOW DOES IT WORK?

Independence describes relationships between random variables, but we use it for function pairs. Which functional relationship does statistical independence translate to? In this part, we argue with proofs and observations that the answer to the above question is *the dissimilarity of invariance types learned by a pair of models*.

**Linear case: Equivalence between independence and orthogonality.** We study the set of function pairs $(f, g)$ satisfying $f(X) \perp\!\!\!\perp g(X)$ for suitable random variable $X \sim p(X)$. Assuming linearity of involved functions and the normality of $X$, we obtain the equivalence between statistical independence and functional orthogonality.

**Lemma 1.** Assume that $f$ and $g$ are affine mappings $f(x) = Ax + a$ and $g(x) = Bx + b$ where $A \in \mathbb{R}^{m \times n}$ and $B \in \mathbb{R}^{l \times n}$. Assume further that $X$ is a normal distribution with mean $\mu$ and covariance matrix $\Sigma$. Then, $f(X) \perp\!\!\!\perp g(X)$ if and only if $\ker(A)^\perp \perp_\Sigma \ker(B)^\perp$. For a positive semi-definite matrix $\Sigma$, we define $\langle r, s \rangle_\Sigma = \langle r, \Sigma s \rangle$, and the set orthogonality $\perp_\Sigma$ likewise. Proof in §A.

In particular, when $f$ and $g$ have 1-dimensional outputs, the independence condition is translated to the orthogonality of their weight vectors and decision boundaries. From a machine learning point of view, $f$ and $g$ are models with orthogonal invariance types.

**Non-linear case: HSIC as a metric learning objective.** We lack theories to fully characterise general, possibly non-linear, function pairs $(f, g)$ achieving $f(X) \perp\!\!\!\perp g(X)$; it is an interesting open question. For now, we make a set of observations in this general case, using the finite-sample independence criterion $\mathrm{HSIC}_0(f, g) := (m-1)^{-2}\mathrm{tr}(\widetilde{f} \, \widetilde{g}^T) = 0$, where $\widetilde{f}$ is the mean-subtracted kernel matrix $\widetilde{f}_{ij} = k(f(x_i), f(x_j)) - m^{-1} \sum_k k(f(x_i), f(x_k))$ and likewise for $\widetilde{g}$ (see §3.1).

Note that $\mathrm{tr}(\widetilde{f} \, \widetilde{g}^T)$ is an inner product between flattened matrices $\widetilde{f}$ and $\widetilde{g}$. We consider the inner product minimising solution for $f$ on an input pair $x_0 \neq x_1$ given a fixed $g$. The problem can be written as $\min_{f(x_0), f(x_1)} \mathrm{tr}(\widetilde{f} \, \widetilde{g}^T)$, which is equivalent to $\min_{f(x_0), f(x_1)} \widetilde{f}_{01} \cdot \widetilde{g}_{10}$.

Suppose $\widetilde{g}_{10} > 0$. It indicates a relative invariance of $g$ on $(x_1, x_0)$, since $k(g(x_1), g(x_0)) > m^{-1} \sum_i k(g(x_1), g(x_i))$. Then, the above problem boils down to $\min_{f(x_0), f(x_1)} \widetilde{f}_{01}$, signifying the relative *variance* of $f$ on $(x_0, x_1)$. Following a similar argument, we obtain the converse statement: if $g$ is relatively variant on a pair of inputs, invariance of $f$ on the pair minimises the objective.

We conclude that $\min_f \mathrm{HSIC}_0(f, g)$ against a fixed $g$ is a metric-learning objective for the embedding $f$, where ground truth pairwise matches and mismatches are relative mismatches and matches for $g$, respectively. As a result, $f$ and $g$ learn different sorts of invariances.

**Effect of HSIC regularisation on toy data.** We have established that HSIC regularisation encourages the difference in model invariances. To see how it helps to de-bias a model, we have prepared

synthetic two-dimensional training data following the cross-domain generalisation case in Figure 1: $X = (B, S) \in \mathbb{R}^2$ and $Y \in \{$red, yellow, green$\}$. Since the training data is perfectly biased, a multi-layer perceptron (MLP) trained on the data only shows 55% accuracy on de-biased test data (see decision boundary figure in Appendix §B). To overcome the bias, we have trained another MLP with equation 3 where the bias-characterising class $G$ is defined as the set of MLPs that take only the bias dimension as input. This model exhibits de-biased decision boundaries (Appendix §B) with improved accuracy of 89% on the de-biased test data.

# 4 EXPERIMENTS

In the previous section, REBI'S has been introduced and theoretically justified. In this section, we present experimental results of REBI'S. We first introduce the setup, including the biases tackled in the experiments, difficulties inherent to the cross-bias evaluation, and the implementation details (§4.1). Results on Biased MNIST (§4.2) and ImageNet (§4.3) are shown afterwards.

## 4.1 EXPERIMENTAL SETUP

**Which biases do we tackle?**    There is a broad spectrum of bias types to be addressed under the cross-bias generalisation setting. Our work is targeting the biases that arise due to the existence of shortcut cues that are sufficient for recognition in training data. In the experiments, we tackle a representative bias of such type: "local pattern" biases for image classification. Even if a CNN image classifier has wide receptive fields, empirical evidence indicates that they heavily rely on local patterns (i.e., color and texture) as opposed to global shape cues (Geirhos et al., 2019).

While it is difficult to precisely define and quantify all local pattern biases, it is easy to capture it through a class of CNN architectures: those with smaller receptive fields. This is precisely the setting where we benefit from REBI'S.

**Evaluating cross-bias generalisation is difficult.**    To measure the performance of a model across real-world biases, one requires an unbiased dataset or one where the types and degrees of biases can be controlled. Unfortunately, data in real world arise with biases. To de-bias a frog and bird image dataset with swamp and sky backgrounds (see §2.1), either rare data samples must be collected (search for photos of a frog on sky) or one must intervene with the data generation process (throw a frog into the sky and take a photo). Either way, it is an expensive procedure (Peyre et al., 2017).

Preparing an unbiased data is feasible in some cases when the bias type is simple (e.g., collecting natural language corpus of unbiased gender pronouns, Webster et al. (2018)). However, we are addressing biases that are expressed in terms of a class of representations but perhaps are difficult to precisely express in language, such as texture bias of image classifiers (§2.3).

We thus evaluate our method along two axes: (1) Biased MNIST and (2) ImageNet. Biased MNIST contains synthetic biases (colour and texture) which we freely control in training and test data for in-depth analysis of REBI'S. In particular, we can measure its performance on perfectly unbiased test data. On ImageNet, we evaluate our method against realistic biases. Due to the difficulty of defining and obtaining bias labels on real images, we use proxy ground truths for the local pattern bias to measure the cross-bias generalisability. MNIST and ImageNet experiments complement each other in terms of experimental control and realism.

**Implementation of REBI'S.**    We describe the specific design choices in REBI'S implementation (equation 3) in our experiments. We will open source the code and data.

To train a model that overcomes local pattern biases, we first define biased model architecture families $G$ such that they precisely and sufficiently encode biased representations: CNNs with relatively small receptive fields (RFs). The biased models in $G$ will by design learn to predict the target class of an image through only local cues. On the other hand, we define a larger search space $F$ with larger RFs for our unbiased representations.

In our work, all networks $f$ and $g$ are fully convolutional networks. $f(x)$ and $g(x)$ denote the final convolutional layer outputs (feature maps), on which we compute the independence measures like

HSIC, CKA, and SCKA (§3.1). We perform global average pooling and a learnable linear classifier on $f(x)$, trained along with the outer optimisation, to compute the cross-entropy loss in equation 3.

For Biased MNIST, $F$ is the `LeNet` $F$ (LeCun et al., 1998) architecture with RF 28. It has two 5-convs[3], after each of which max-pooling layers of $2 \times 2$ kernels are applied, followed by three linear layers. $G$ has the same number of layers but either with all convolutional layers of $1 \times 1$ kernels and without max-pooling operations, called `BlindNet1`, or with one 3-conv and one 1-conv, called `BlindNet3`, each with a RFs of 1 and 3, respectively. On ImageNet, we use `ResNet` (He et al., 2016) architecture for $F$ (either `ResNet18` with RF of 435 or `ResNet50` with RF of 427). $G$ is defined as `BagNet` (Brendel & Bethge, 2019) architectures with the same depth as `ResNet`'s (either `BagNet18` with RF of 43 or `BagNet50` with RF of 91). More implementation details are provided in §C.

## 4.2    BIASED MNIST

We first verify our model on a dataset where we have full control over the type and amount of bias during training and evaluation. We describe the dataset and present the experimental results.

### 4.2.1    DATASET AND EVALUATION

We construct a new dataset called **Biased MNIST** designed to measure the extent to which models generalise to bias shift. We modify MNIST (LeCun et al., 1998) by introducing two types of bias $B$ – colour and texture – that highly correlate with the label $Y$ during training. With $B$ alone, a CNN can achieve high accuracy without

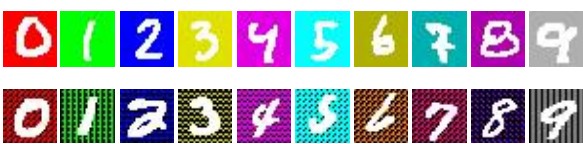

**Figure 2: Biased MNIST.** We construct a synthetic dataset with two types of biases – colour and texture – which highly correlate with the label during training. Upper row: colour bias. Lower row: colour and texture biases.

having to learn inherent signals for digit recognition $S$, such as shape, providing little motivation for the model to learn beyond these superficial cues.

We inject colour and texture biases by adding colour or texture patterns on training image backgrounds (see Figure 2). We pre-select 10 distinct colour or $3 \times 3$ texture patterns $b(y)$ for each digit $y \in \{0, \cdots, 9\}$. Then, for each image of digit $y$, we assign the pre-defined pattern $b(y)$ with probability $\rho \in [0, 1]$ and any other pattern (pre-defined for other digits) with probability $(1 - \rho)$. $\rho$ then controls the bias-target correlation in the training data: $\rho = 1.0$ leads to complete bias and $\rho = 0.1$ leads to an unbiased dataset. We consider two datasets: **Single-bias MNIST** with only colour bias and **Multi-bias MNIST** with both colour and texture biases ($\rho = 0.99$ in all the experiments).

We evaluate the models generalisability to bias shift by evaluating under the following criterion:

**Biased.**    $p(S^{\text{te}}, B^{\text{te}}, Y^{\text{te}}) = p(S^{\text{tr}}, B^{\text{tr}}, Y^{\text{tr}})$ (an in-distribution case in §2.1). Whatever bias the training set contains, it is replicated in the test set. This measures the ability of de-biased models to maintain high in-distribution performance while generalising to unbiased settings.

**Unbiased.**    $B^{\text{te}} \perp\!\!\!\perp Y^{\text{te}}$. We assign biases on test images independently of the labels. Bias is no longer predictive of $Y$ and a model needs to utilise actual signals $S$ to yield correct predictions.

We have additional fine-grained measures on Multi-bias MNIST: removed colour bias (**colour**; texture bias remains) and removed texture bias (**texture**; colour bias remains) cases. Colour and texture biases are marginalised out in the test set, respectively, to factorise generalisability across different types of bias.

### 4.2.2    RESULTS

Results on Single- and Multi-bias MNIST are shown in Table 1.

**REBI'S lets a model overcome bias.**    We observe that vanilla `LeNet` $F$ achieves 100% accuracy under the "biased" metric (the same bias between training and test data) in Single- and Multi-bias MNIST. This is how most machine learning tasks are evaluated, yet this does not show the extent

---

[3]We refer to convolutional layers with $k \times k$ kernels as $k$-conv.

| Model | Description | Single-bias MNIST | | Multi-bias MNIST | | | |
|---|---|---|---|---|---|---|---|
| | | biased | unbiased | biased | colour | texture | unbiased |
| $F$ | Vanilla | **100.** | 27.5 | **100.** | 45.1 | 85.8 | 29.6 |
| $G$ | Biased | **100.** | 10.8 | **100.** | 10.0 | 58.5 | 4.7 |
| $F_{\text{HEX}}$ | Wang et al. (2019) | 97.2 | 14.7 | 95.0 | 31.1 | **91.1** | 18.1 |
| $F_{\perp\!\!\!\perp G}$ | REBI'S (ours) | 96.1 | **74.9** | 94.7 | **91.9** | 90.9 | **88.6** |

**Table 1: Biased MNIST results.** Architecture families are set as $F = \texttt{LeNet}$ and $G = \texttt{BlindNet1}$ for Single-bias and $\texttt{BlindNet3}$ for Multi-bias MNIST. Accuracy results are shown.

to which the model depends on bias for prediction. When the bias cues are randomly assigned to the label at evaluation, vanilla $\texttt{LeNet}$ accuracy collapses to 27.5% and 29.6% under the "unbiased" metric on Single- and Multi-bias MNIST, respectively. The intentionally biased $\texttt{BlindNet}$ models $G$ result in an even lower accuracy of 10.8% on Single-bias MNIST, close to the random chance 10%. This reveals that the seemingly high-performing model has in fact overfitted to bias and has not learned beyond this fallible strategy.

$\texttt{REBI'S}$, on the other hand, achieves robust generalisation across all settings by learning to be different from $\texttt{BlindNet}$ representations $G$. $\texttt{REBI'S}$ achieves a +47.4 pp (Single) and +59.0 pp (Multi) higher performances than the vanilla model under the cross-bias generalisation setup (the unbiased metric), with a slight degradation in original accuracies (-3.9 pp and -5.3 pp, respectively).

**Comparison against HEX.** Previously, $\texttt{HEX}$ (Wang et al., 2019) has attempted to reduce the dependency of a model on "superficial statistics", or high-frequency textural information. $\texttt{HEX}$ measures texture via neural grey-level co-occurrence matrices (NGLCM) and projects out the NGLCM feature from the output of the model of interest. We observe that $F_{\text{HEX}}$, where $\texttt{HEX}$ is applied on $F$, is effective in removing texture biases (85.8% to 91.1% cross-texture accuracy), but still vulnerable to colour biases (accuracy drops from 45.1% to 31.1%). Hand-crafting texture features as done by $\texttt{HEX}$ has resulted in its limited applicability beyond the hand-crafted bias type. By designing the model family architecture, instead of a specific feature extractor, $\texttt{REBI'S}$ achieved a representation free of broader types of biases.

### 4.2.3 FACTOR ANALYSIS

Some design choices exist, leading to our final model (§3.1). We examine how the factors contribute to the final performance. See Table 2 for ablative studies on the Single-bias MNIST.

**Impact of the independence criterion.** Three independence measures, HSIC, CKA, and SCKA, have been considered. SCKA, the separation CKA, used in $\texttt{REBI'S}$, shows a superior de-biasing performance (74.9%) against baseline choices. It improves upon HSIC by avoiding the trivial solution (a constant function), and upon CKA via more stable optimisation due to the milder conditional independence $f(X) \perp\!\!\!\perp B \,|\, Y$ that does not contradict the classification objective.

| Indep. | biased | unbiased |
|---|---|---|
| HSIC | 83.5 | 51.1 |
| CKA | 97.4 | 22.0 |
| **SCKA** | **96.1** | **74.9** |

| Inner opt. | biased | unbiased |
|---|---|---|
| SCKA | 95.8 | 62.4 |
| $L_2$ | **96.1** | **74.9** |

| Updating $g$ | biased | unbiased |
|---|---|---|
| Fixed $g$ | 94.6 | 44.7 |
| Multiple $g$ | 95.9 | 58.1 |
| **Updated $g$** | **96.1** | **74.9** |

**Impact of the $L_2$ objective in the inner optimisation.** We then study the effect of our choice to replace the inner SCKA optimisation with the $L_2$ objective. $L_2$ is considered, as it poses a stronger convergence condition than SCKA does. We confirm indeed that the $L_2$ objective results in a better de-biasing performance.

**Table 2: Factor analysis.** Default $\texttt{REBI'S}$ parameters: last rows.

**Impact of updating $g \in G$.** The advantage of specifying a class of models $G$ instead of a single, fixed model $g_0$ is that $\texttt{HSIC}(f, G)$ can be computed more precisely (§3.1). We quantify this benefit. Fixing the biased representation to $g_0$ results in sub-optimal de-biasing performance, 44.7%. By including multiple fixed biased models $\{g_0, \cdots, g_7\} \subset G$, de-biasing improves to 58.1%, but is not as good as the updated $g$ case, 74.9%. It is thus important to precisely compute the representation-to-set independence. More detailed analysis around the receptive field sizes of models in $G$ is in Appendix §E.

| Model | Description | Biased | Unbiased | IN-A |
|---|---|---|---|---|
| $F$ | ResNet18 | 93.3 | 85.8 | 30.5 |
| $G$ | BagNet18 | 72.4 | 58.6 | 19.5 |
| $F_{\perp\!\!\!\perp G}$ | REBI'S | **93.7** | **88.4** | **31.7** |
| $F$ | Geirhos et al. | 92.5 | 87.6 | 29.7 |
| $F$ | ResNet50 | **91.7** | 78.3 | 29.5 |
| $G$ | BagNet50 | 73.0 | 60.9 | 21.4 |
| $F_{\perp\!\!\!\perp G}$ | REBI'S | 88.7 | **89.2** | **31.3** |

**Table 3: ImageNet results.** We show results with $F$ and $F_{\perp\!\!\!\perp G}$ corresponding to ($F$ is ResNet18 and $G$ is BagNet18) and ($F$ is ResNet50 and $G$ is BagNet50). IN-A indicates ImageNet-A.

## 4.3 ImageNet

In ImageNet experiments, we further validate the applicability of REBI'S on the local pattern bias in realistic images (i.e., objects in natural scenes). The local pattern bias often lets a model achieve good in-distribution performances by exploiting the local cue shortcuts (e.g, determining a turtle class by not seeing its shape but the background texture).

### 4.3.1 Dataset and evaluation

We construct **9-Class ImageNet**, a subset of ImageNet (Russakovsky et al., 2015) containing 9 super-classes as done in Ilyas et al. (2019), since full-scale analysis on ImageNet is not scalable. We additionally balance the ratios of sub-class images for each super-class to solely focus on the effect of the local pattern bias.

Since it is difficult to evaluate cross-bias generalisability on realistic unbiased data (§4.1), we settle for alternative evaluations:

**Biased.** $p(S^{\text{te}}, B^{\text{te}}, Y^{\text{te}}) = p(S^{\text{tr}}, B^{\text{tr}}, Y^{\text{tr}})$ Accuracy is measured on the in-distribution validation set. Though widely-used, this metric is blind to a model's generalisability to unseen bias-target combinations.

**Unbiased.** $B^{\text{te}} \perp\!\!\!\perp Y^{\text{te}}$ As a proxy to the perfectly de-biased test data, which is difficult to collect (§4.1), we use *texture* clusters IDs $c \in \{1, \cdots, K\}$ as the ground truth labels for local pattern bias. For full details of texture clustering algorithm, see Appendix §F. For an unbiased accuracy measurement, we compute accuracies for every set of images corresponding to a target-texture combination $(c, y)$. The combination-wise accuracy $A_{c,y}$ is computed by $\text{Corr}(c, y)/\text{Pop}(c, y)$, where $\text{Corr}(c, y)$ is the number of correctly predicted samples in $(c, y)$ and $\text{Pop}(c, y)$ is the total number of samples in $(c, y)$, called the **population** at $(c, y)$. The unbiased accuracy is then the mean accuracy over all $A_{c,y}$ where the population is non-zero $\text{Pop}(c, y) \neq 0$. This measure gives more weights on samples of unusual texture-class combinations (smaller $\text{Pop}(c, y)$) that are less represented in the usual biased accuracies. Under this unbiased metric, a biased model basing its recognition on textures is likely to show sub-optimal results on unusual combinations, leading to a drop in the unbiased accuracy.

**ImageNet-A.** ImageNet-A (Hendrycks et al., 2019) contains failure cases of ImageNet pre-trained ResNet50 among web images. The images consist of many failure modes of networks when "frequently appearing background elements" become erroneous cues for recognition (e.g. a bee image feeding on hummingbird feeder is recognised as a hummingbird). Improved performance on ImageNet-A is an indirect signal that the model learns beyond the bias shortcuts.

### 4.3.2 Results

We measure performances of ResNet18 and ResNet50, each trained to be different from BagNet18 and BagNet50 respectively using REBI'S, under the metrics in the previous part. Results are shown in Table 3.

**Vanilla models are biased.** Both ResNet18 and ResNet50 show good performances on the biased accuracy (93.3% and 91.7%, respectively), but dropped performances on the texture-unbiased accuracies (85.8% and 78.3%, respectively). BagNet18 and BagNet50 perform worse than the vanilla ResNets as they are heavily biased towards texture by design (i.e., small receptive field size). The drop signifies the biases of vanilla models towards texture cues; by basing their predictions

on texture cues they obtain generally better accuracies on texture-class pairs $(c, y)$ that are represented more. The drop also shows the limitation of current evaluation schemes where cross-bias generalisation is not measured.

**REBI'S leads to less biased models.** When `REBI'S` is applied on `ResNet18` and `ResNet50` to encourage them to unlearn cues learnable by `BagNet18` and `BagNet50`, respectively, we observe general boost in unbiased accuracies. `ResNet18` improves from 85.8% to 88.4%; `ResNet50` from 78.3% to 89.2%. Our method thus robustly generalises to less represented texture-target combinations at test time. We observe that our method also shows improvement on the challenging ImageNet-A subset (e.g. from 29.5% to 31.3% for `ResNet50`), which further shows our superiority on generalisability to unusual texture-class combinations. Similarly, data augmentation via Stylised ImageNet (Geirhos et al., 2019) attempts to reduce a model's dependency on texture by augmenting data with texturised images. While it shows improvements in reducing texture bias (87.6% for unbiased accuracy), it does not increase the generalisability to the challenging natural adversarial examples (29.7% for ImageNet-A). More detailed analysis on per-texture and per-class accuracies are included in Appendix §G; learning curves for the baseline and `REBI'S` are in Appendix §H.

**Qualitative analysis.** We qualitatively present the cases where our method successfully de-biases a texture-target dependency. Figure 3 shows examples of common and uncommon texture-target combinations for "grass" and "close-up" texture clusters. The shown uncommon instances are the ones `ResNet18` has incorrectly predicted the class. For example, crab in the grass has been predicted as turtle, presumably because turtles co-occur a lot with grass backgrounds in training data. On the other hand, `REBI'S` (ours) robustly generalises to unusual texture-class combinations.

## 5 CONCLUSION

We have identified a practical problem faced by many machine learning algorithms that the learned models exploit bias shortcuts to recognise the target (cross-bias generalisation problem in §2). In particular, models tend to under-utilise its capacity to extract non-bias signals (e.g. global shapes for object recognition) when bias shortcuts provide sufficient cues for recognition in the training data (e.g. local patterns and background cues for object recognition) (Geirhos et al., 2019). We have addressed this problem with the `REBI'S` framework, which does not rely on expensive, if not infeasible, training data de-biasing schemes. Given an identified set of models $G$ that encodes the bias to be removed, `REBI'S` encourages a model $f$ to be statistically independent of $G$ (§3). We have provided theoretical justifications for the use of statistical independence in §3.2, and have validated the superiority of `REBI'S` in removing biases from models through experiments on modified MNIST and ImageNet (§4).

common images | uncommon images

turtle in grass
baseline: turtle
ours: turtle

crab in grass
baseline: turtle
ours: crab

insect close-up
baseline: insect
ours: insect

frog on close-up
baseline: insect
ours: frog

Figure 3: **Qualitative results.** Common and uncommon images are shown according to class-texture relationship. Predictions of `ResNet18` and `REBI'S` are shown as well.

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

## A    STATISTICAL INDEPENDENCE IS EQUIVALENT TO FUNCTIONAL ORTHOGONALITY FOR LINEAR MAPS.

We provide a proof for the following lemma in §3.2.

**Lemma 1.**    Assume that $f$ and $g$ are affine mappings $f(x) = Ax + a$ and $g(x) = Bx + b$ where $A \in \mathbb{R}^{m \times n}$ and $B \in \mathbb{R}^{l \times n}$. Assume further that $X$ is a normal distribution with mean $\mu$ and covariance matrix $\Sigma$. Then, $f(X) \perp\!\!\!\perp g(X)$ if and only if $\ker(A)^\perp \perp_\Sigma \ker(B)^\perp$. For a positive semi-definite matrix $\Sigma$, we define $\langle r, s \rangle_\Sigma = \langle r, \Sigma s \rangle$, and the set orthogonality $\perp_\Sigma$ likewise.

**Proof.**    Due to linearity and normality, the independence $f(X) \perp\!\!\!\perp g(X)$ is equivalent to the co-variance condition $\text{Cov}(f(X), g(X)) = 0$. Covariance is computed as:

$$\text{Cov}(f(X), g(X)) = \mathbb{E}_X A(X - x^0)(X - x^0)^T B^T = A\Sigma B^T \tag{4}$$

Note that

$$A\Sigma B^T = 0 \iff \langle v, A\Sigma B^T w \rangle = 0 \,\forall\, v, w \iff \langle A^T v, B^T w \rangle_\Sigma = 0 \,\forall\, v, w \tag{5}$$
$$\iff \text{im}(A^T) \perp_\Sigma \text{im}(B^T) \iff \ker(A)^\perp \perp_\Sigma \ker(B)^\perp \quad \square$$

## B    DECISION BOUNDARY VISUALISATION FOR TOY EXPERIMENT

We show the decision boundaries of the toy experiment in §3.2. See Figure 4. GIF animations of decision boundary changes over training can be found at anonymous link.

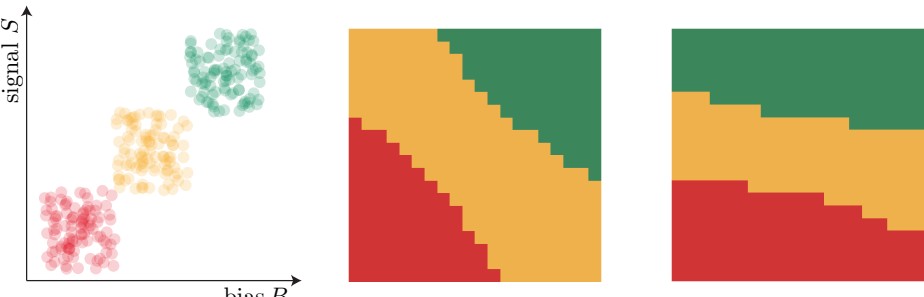

Figure 4: **Decision boundaries**. Left to right: training data, baseline model, and our model.

## C    IMPLEMENTATION DETAILS

We solve the minimax problem in equation 3 through alternating stochastic gradient descents (ADAM Kingma & Ba (2014)), where we alternate between 5 epochs for the outer problem and 5 epochs for the inner one. The regularisation parameter $\lambda$ is set to $7,000$ for CKA and $100$ for $L_2$ minimisation on MNIST and $0.05$ for ImageNet. Note that we use a large $\lambda$ for MNIST as the degree of synthetic bias is excessively large (bias-target correlation set to $\rho = 0.99$) compared to the degree of bias in realistic settings. We have used batch sizes $256$ ($32$ for `ResNet50`) and learning rates $0.001$ with linear decay. We train all models for $80$ epochs for Biased MNIST. For 9-class ImageNet, we train `ResNet18` for $200$ epochs and `ResNet50` for $100$ epochs.

`BagNet18` and `BagNet50` have $18$ and $50$ convolutional layers which are the same structure (Basic or Bottleneck blocks) as `ResNet18` and `ResNet50`, respectively. The internal kernel sizes of `BagNets` are set to $1 \times 1$ following the original paper's philosophy (Brendel & Bethge (2019)).

## D    PERFORMANCE BY BIAS AND LABEL ON BIASED MNIST.

In the Biased MNIST experiments (§4.2), we have shown either biased or unbiased set statistics. In this section, we provide more detailed results where model accuracies are computed per bias class

$B = b$ and per target label $Y = y$. We visualise the case-wise accuracies of the baseline `LeNet` and the `REBI'S` trained `LeNet`$_{\perp\!\!\!\perp \text{BlindNet}}$ in Figure 5. The diagonals in each matrix indicate the pre-defined bias-target pair (§4.2.1). Thus, the biased accuracies can be computed by taking the mean of diagonal entries in each matrix and the unbiased accuracies through the mean of all entries in each matrix.

The vanilla `LeNet`'s tendency to have higher accuracies on diagonal entries and near-zero performances on many other off-diagonal entries indicate the fact that `LeNet` is relying a lot on the colour (bias) cues. `REBI'S` successfully resolves this tendency in the vanilla model, exhibiting more uniform performances across all bias-target pairs $(b, y)$. Note that accuracies below the main diagonal are relatively high as they are the classes that pre-defined patterns are assigned to at probability $(1 - \rho)$. Figure 6 demonstrates that our method is successful across different degrees of bias in a given training data.

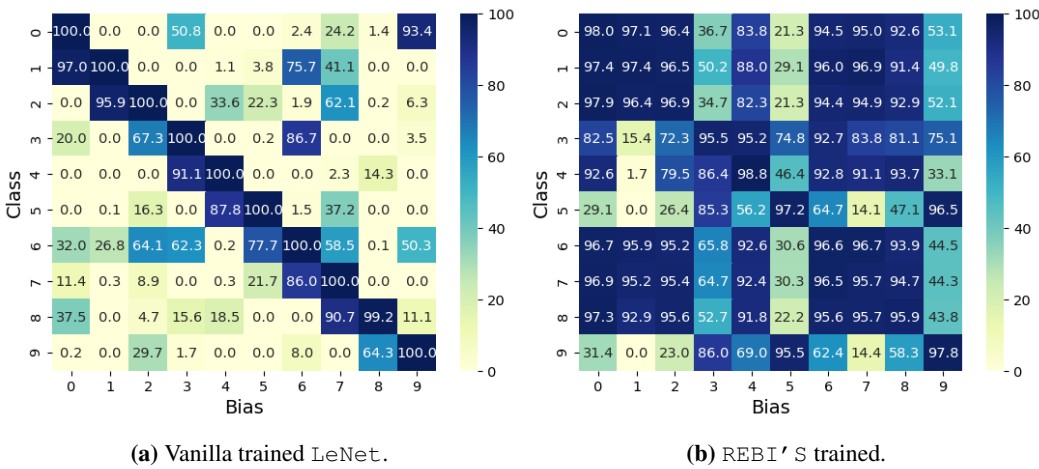

(a) Vanilla trained `LeNet`.  (b) `REBI'S` trained.

**Figure 5: Bias-target-wise accuracies.** We show accuracies for each bias $B = b$ and target $Y = y$ pair in Single-bias MNIST.

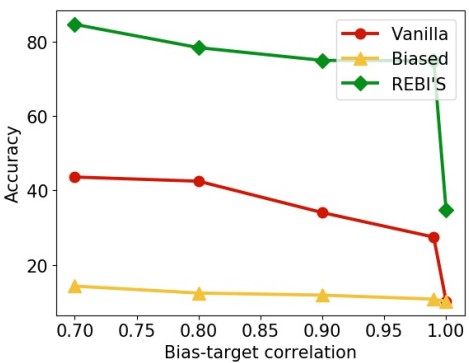

**Figure 6: Impact of $\rho$.** `REBI'S` is effective in de-biasing across different degree of bias in data.

# E    IMPACT OF RECEPTIVE FIELDS OF $G$ FOR BIASED MNIST

It is conceptually important to design a the set of biased models $G$ that encode the bias $B$ as precisely as possible. See precision and recall conditional for $G$ in §2.3). To see if this is empirically true, we have measured the performance of `REBI'S` with $F = $ `LeNet` and $G = $ `BlindNet`, where the `BlindNet` receptive fields are controlled by replacing convolutional layers with $1 \times 1$ convolutions, resulting in receptive fields $\{1, 5, 7, 28\}$. The $28 \times 28$ receptive field indicate the case when `LeNet` is used as $G$. We measure the biased and unbiased set performances on Single-bias MNIST (§4.2.1).

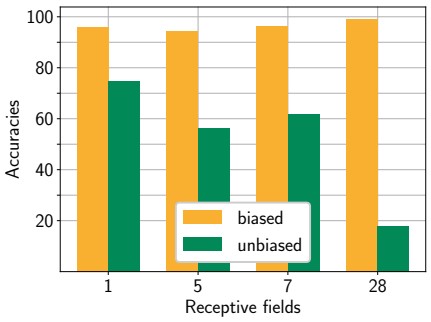

**Figure 7: Receptive fields of** $G$**.** Biased and unbiased accuracies of `REBI'S` with $F = $ `LeNet` and $G = $ `BlindNet` with varying receptive fields.

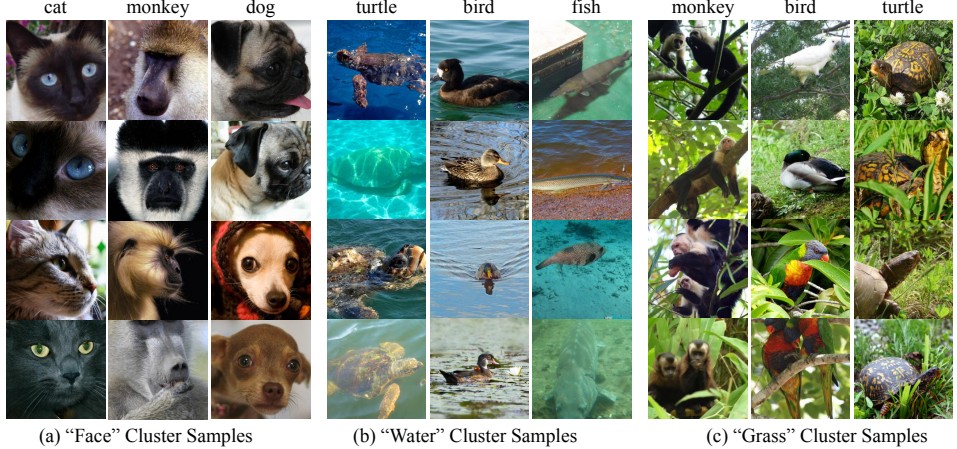

   (a) "Face" Cluster Samples       (b) "Water" Cluster Samples       (c) "Grass" Cluster Samples

**Figure 8: Texture-class correlation.** We show samples from each texture cluster. For each cluster, we visualise its top-3 correlated classes in rows.

Results are shown in Figure 7. We observe relatively stable biased set performances and decreasing unbiased set performances. The decrease in de-biasing ability is attributed to the violation of the precision condition for $G$ (§2.3): $1 \times 1$ receptive fields are sufficient to capture any colour bias variations, and de-biasing against models of larger receptive fields will further make $f$ not see the meaningful signal $S$ (cues beyond $1 \times 1$-expressible colours). In the extreme case, when $F = $ `LeNet` is trained to be independent against itself $G = $ `LeNet`, the de-biasing performance drops significantly.

## F    TEXTURE CLUSTERING

In our ImageNet experiments (§4.3), we obtained proxy ground truths for the local pattern bias using texture clustering. We extract texture information from images by clustering the gram matrices of low-layer feature maps as done in standard texturisation methods (Gatys et al., 2015; Johnson et al., 2016); we use feature maps from layer `relu1_2` of a pre-trained `VGG16` (Simonyan & Zisserman, 2014). As we intend to evaluate whether a given model is biased towards local pattern cues, we only utilise features from the lower layer encoding lower layer features like edges and colours rather than high-level semantics. Figure 8 shows that each cluster effectively captures similar local patterns across different classes. For each cluster, we visualise its top-3 correlated classes. We can see that certain classes share a common texture: cat, monkey, and dog share a "face"-like texture. If a certain class is biased towards a particular texture during training, a model can take the shortcut to utilise texture cues for recognising the target class, leading to a sub-optimal cross-bias generalisation to unusual class-texture combinations (e.g. crab on grass).

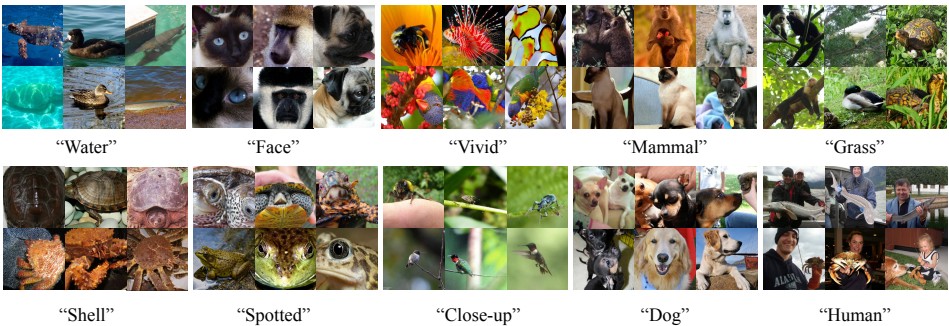

"Water"     "Face"     "Vivid"     "Mammal"     "Grass"

"Shell"     "Spotted"     "Close-up"     "Dog"     "Human"

**Figure 9: More clustering samples.** Extended version of Figure 8.

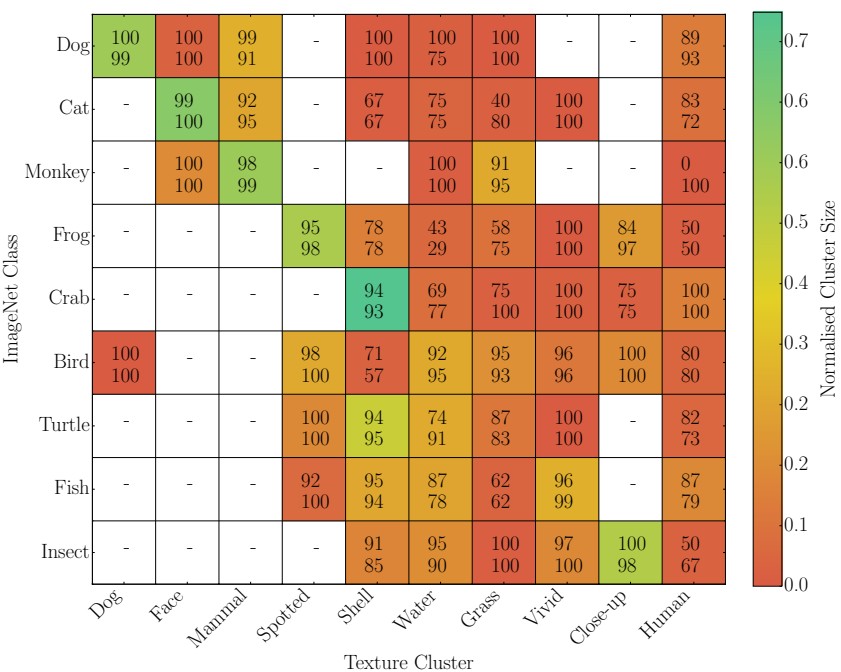

**Figure 10: Data and model bias.** Rows correspond to image labels and columns correspond to texture clusters. Cells are colour-coded according to the population of samples of corresponding texture-class pair. In each cell, `ResNet18` and `REBI'S` accuracies are shown in pairs.

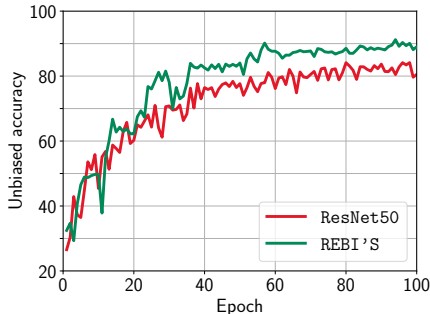

**Figure 11: Learning curve.** De-biased ImageNet accuracies of vanilla `ResNet50` and `REBI'S` trained against `BagNet50`.

## G  BIAS IN DATA AND MODELS

We study the bias in the 9-class ImageNet data and the models trained on them (§4.3). Figure 10 shows the statistics of texture biases in data and the model biases that result from them. To measure the dataset bias, we empirically observe correlations between texture and target classes by counting the number of samples for each texture-class pair $(c, y)$ (denoted "population" $\text{Pop}(c, y)$ in the main paper). We observe indeed that there does exist a strong correlation between texture clusters and class labels. We say that a class has a *dominant texture cluster* if the largest cluster for the class contains more than half of the class samples. 6 out of 9 classes considered has the dominant texture cluster: ("Dog", "Dog"), ("Cat", "Face"), ("Monkey", "Mammal"), ("Frog", "Spotted"), ("Crab", "Shell") and ("Insect", "Close-up").

Figure 10 further shows the accuracies of the baseline `ResNet18` and `REBI'S` to indicate the presence of bias in the models, and how `REBI'S` overcomes the bias despite the bias in data itself. We measure the average of accuracies in classes with dominant texture clusters (biased classes) and the average in less biased classes. We observe that `ResNet18` shows higher accuracy on biased classes (89.4%) than on less biased classes (83.9), signifying its bias towards texture. On the other hand, `REBI'S` achieves similar accuracies 88.7%(biased classes) and 88.2% (unbiased classes). We stress that `REBI'S` overcomes the bias even if the training data itself is biased.

## H  LEARNING CURVES ON IMAGENET

We visualise the learning curves for the baseline vanilla `ResNet50` and `REBI'S` trained `ResNet50` against `BagNet50` in Figure 11. We observe that `REBI'S` gradually de-biases a representation beyond the baseline model.

