# OpenReview forum: "Learning De-biased Representations with Biased Representations"
_ICLR.cc/2020/Conference — Reject_

### Official Review · AnonReviewer3 · 2019-10-22
**Official Blind Review #3**

**Rating:** 6

**Review:**

###  Summary

The paper proposes a method for regularizing neural networks to mitigate certain known biases from the representations learned by CNNs. The authors look at the setting in which the distribution of biases in the train and test set remains the same, but the distribution of targets given biases changes.

To remove the bias, they propose hand-designing a set of models that rely exclusively on these biases to make predictions. They then train a new unbiased model by making sure that this new model uses sufficiently different information than the biased models to make predictions (By adding a regularization term)

### Decision and reasons

I vote for a weak accept.

Strengths:
1: The paper is well written with a clearly defined problem-setting. The proposed method is sound and interesting, and the empirical results, thorough.

Weaknesses:
2: The solution just pushes the problem of 'learning a model that only uses the true signal to make predictions' to 'learning a set of models that only use the noise to make predictions.' It's not clear why the latter is easier than the former.

3: The paper does not have any baselines that directly try to remove the bias (Instead of using the two-step process). As a result, it's hard to judge how meaningful the improvements are.


### Supporting arguments for the reasons for the decision.

Strengths:
1: The paper does a really good job of defining the problem-setting. Contrasting cross-bias with cross-domain and in-distribution makes the goal of the paper very clear. The notation used to formalize the problem setting in Section 2.1 is also clear and concise. Moreover, the experiments on the toy dataset help clarify the proposed solution whereas experiments on Biased MNIST and Imagenet show that it successfully mitigates the bias. Finally, the authors show the importance of each component of the proposed solution by factor analysis.

Weaknesses:
2: In the most general case, it is not obvious why it is easier to define and learn a set of models that only use noise to make predictions (which is the required first step for their proposed solution) as opposed to learning a model that only uses signal (which is the goal of the problem). These two problems seem equally hard. The paper builds on the premise that in some cases former is easier (i.e. in some cases, it is easier to learn a set of models that use only noise as opposed to learning a debiased model directly). The authors give two such examples (They only explore the first experimentally.)
1. Learning a model that relies on local texture. They achieve this by limiting the receptive field of the features.
2. Learning a model that relies on static images to make predictions about actions in videos.
I feel that the two given examples are very narrow. It would be nice if the authors could identify a broader class of problems for which G is given or can easily be defined. Moreover, even in these two examples, I'm not convinced that the proposed biased models only use B for making predictions. For example, for some classification tasks, the local texture could be part of the signal and not just the bias. Similarly, static images from a video do contain important information for making the prediction.

3: The authors only compare their method to a baseline that does nothing to debias the representations. Even though this is an important comparison (as it shows that the proposed method can debias the representations), it does not tell the reader how effective the proposed method is compared to other possible solutions. The results would be more meaningful if the authors could include at-least a simple baseline that tries to remove the bias in other ways (For example, they could use the style-transfer baseline used by Geirhos et al., 2019).

I vote for accepting the paper as a poster. It introduces an interesting approach for debiasing representations. However, due to its narrow scope and missing baselines, I would not recommend the paper for an oral presentation.


### Questions

1. What are some broader class of problems for which defining G is easier than directly regularizing for debiased representations?

2. How well do Bagnets alone perform on the benchmarks in Table 3? I would expect to see that Bagnets alone do worse than vanilla Resnets on Unbiased and IN-A. Is that so?

3. How well do other methods do in these domains? (Such as methods that directly debias the training data against texture by applying style-transfer).


### Update after Author's response

The author's response has clarified the motivation behind the proposed approach to an extent. They have also added a comparison with a method that directly promotes learning the shape as opposed to the texture ( by training on stylized Imagenet)

I agree with R1 on all accounts (i.e. it is very hard to define the family of biased feature extractors, the proposed approach is ad-hoc, the authors need to compare to texture-shape disentanglement methods, etc), however at the same time, I can see that proposed approach can act as a useful heuristic for regularizing neural networks to pay attention to certain kind of information.

An interesting use-case of the proposed method (which the authors indirectly mentioned in their response to my review) is in a multi-modal setting. It's not trivial to enforce deep learning systems to utilize all data modalities in a multi-modal setting. By defining G to be models trained on individual modalities, it would be possible to nudge our models to pay attention to the information in all modalities.

Some important results in deep learning have been ad-hoc (For example skip connections in deep networks, ReLUs) and have nonetheless progressed the field. This work is not as widely applicable as skip connections or ReLUs, but it is, nonetheless, providing a heuristic for solving an important problem.

**Experience Assessment:**

I have read many papers in this area.

**Review Assessment: Checking Correctness Of Derivations And Theory:**

I did not assess the derivations or theory.

**Review Assessment: Checking Correctness Of Experiments:**

I carefully checked the experiments.

**Review Assessment: Thoroughness In Paper Reading:**

I read the paper thoroughly.

---

> ### Author Response · Authors · 2019-11-15
> **Response to Reviewer3 (1/2)**
>
> We thank all the reviewers for their recognition of the task “interesting and important” (R1), and finding the proposed method “sound and interesting” (R2, R3). In particular, R3 has commented that “the paper does a really good job of defining the problem setting”. Yet, we find some lack of detail and clarity in our paper could potentially have led to misunderstanding our work -- we have clarified them in reviewer-specific responses and have updated the paper accordingly.
>
> We summarize our revisions to the paper as follows:
>
> Section 2:
> Based on the helpful comments of R1 and R3, we have clarified that
> * G may capture important signals as well as biases (Section 2.3, paragraph 3).
> * G can be easily defined in a broad class of problems in machine learning applications  (Section 2.3, paragraph 4).
>
> * We have updated additional related works provided by R2 (Section 2.3, paragraph 4).
>
> Section 3:
> * We have clarified that REBI’S does not suppress the signal captured by G due to its formulation as conditional independence (Section 3.1, Independence versus separation).
>
> Section 4:
> * As requested by R3, we have added additional comparison on vanilla BagNets and Stylized ImageNet augmentation [1] on ImageNet (Table 3).
> * There was a minor error in our evaluation code on ImageNet-A, and we have updated the correct values for every method (Table 3).
>
> * We have fixed typos throughout the paper.
>
> Our response to Reviewer 3 is as follows:
>
> #1 I'm not convinced that the proposed biased models (G) only use B for making predictions.
>
> > We agree with R3 that it is difficult to define G that only use B for making predictions (perfect precision, see Section 2.3). Nevertheless, G does not have to be perfect; please also see our response to R1. Our framework does not remove signals captured by G because REBI’S is built upon the “conditional independence” criterion (Section 3.1, “Independence versus separation” paragraph) that f(x) is encouraged to be independent of g(x), **given** the target task Y. Unlike independence that requires f(x) to ignore g(x) altogether, conditional independence allows f(x) to still retain cues captured by g(x) if they highly correlate with Y. In short, our goal is to **discourage over-reliance** towards cues captured by g(x) and push the model to learn other features that are useful for prediction. In the revision, we have improved the description of the precision and recall conditions for G in Section 2.3: those conditions are not meant as hard requirements but as desiderata. Even if they are not precisely met, conditional independence will do the rest of the job.
>
> #2 The solution just pushes the problem of 'learning a model that only uses the true signal to make predictions' to 'learning a set of models that only use the noise to make predictions.' It's not clear why the latter is easier than the former.
>
> > As R3 suggested, there can be problems where directly using the true signal is easier than only using bias for prediction. If that is the case, then we would suggest to use the signals directly. Our work is concerned with the cases where i) the signal is highly entangled with the bias and there is no easy method to disentangle them yet, and ii) we have some evidence for the type of bias and the corresponding set of models G capturing it. We will give examples of such scenarios in #3.
>
> #3 Identify a broader class of problems for which G is given or can easily be defined.
>
> >  As addressed in #1 (and in response to R1), G does not have to be perfect and we are provided certain evidence for bias types and corresponding set of representations G in many application scenarios [2-9]. Action recognition models have been reported to rely heavily on static cues without learning temporal cues [2,3]; though the actual bias may not precisely be _any static cue_, we can still regularize the 3D convolutional networks towards better generalization across static cue biases by defining G to be the set of 2D convolutional architectures. It has been argued that visual question answering (VQA) models, too, rely overly on language biases rather than the visual cues (e.g. without looking at the image, one knows the answer to “what color is the banana” is “yellow”) [4]. We can define G as the set of models looking at the language modality only. Entailment models are biased towards the presence of certain words (e.g. when there are many “not”s, the sentence is “contradictory”), rather than really understanding the underlying meaning of sentences [5,6]. We can design G to be the set of bag-of-words classifiers [7,8]. In the revised paper, we have supplemented the above examples in Section 2.3.

---

> > ### Author Response · Authors · 2019-11-15
> > **Response to Reviewer3 (2/2)**
> >
> >
> > #4 How well do Bagnets alone perform on the benchmarks in Table 3?
> >
> > > Please see the table below.
> >
> > Models                                      	Biased           	Unbiased       IN-A
> > ResNet18                                 	93.3             	85.8                	30.5
> > BagNet18                                 	72.4                	58.6                	19.5
> > ResNet18-BagNet18                	93.7                	88.4                	31.7
> >
> > Models                                      	Biased           	Unbiased       IN-A
> > ResNet50                                 	91.7                	78.3                	29.5
> > BagNet50                                 	73.0                	60.9                	21.4
> > ResNet50-BagNet50                	88.7                	89.2               	31.3
> >
> > BagNets alone perform worse than the vanilla ResNets themselves as they are biased towards texture by design (i.e., small receptive field size). We have updated the results in the paper (Table 3).
> >
> > #5 How well do other methods do in these domains (Unbiased and IN-A)?
> >
> > > As requested by R3, we compare REBI’S against the data augmentation strategy using Stylized ImageNet [9].
> >
> > Models                                                   	Biased           	Unbiased       IN-A
> > ResNet18                                           	        93.3                	85.8                	30.5
> > StylizedImageNet ResNet18                 	92.5                	87.6                	29.7
> > REBI’S (ResNet18-BagNet18)	        	93.7                	88.4                	31.7
> >
> > Stylized ImageNet augmentation shows improvements in reducing texture bias (“unbiased” accuracy from 85.8 to 87.6), but it does not increase the generalizability to the challenging natural adversarial examples (ImageNet-A accuracy from 30.5 to 29.7). REBI’S improves upon the StylizedImageNet-trained model both in terms of the unbiased and ImageNet-A accuracies (1.2 pp and 2.0 pp, respectively). We have updated the results in Table 3.
> >
> > <References>
> > [1] https://arxiv.org/abs/1610.02413
> > [2] http://openaccess.thecvf.com/content_ECCV_2018/papers/Yingwei_Li_RESOUND_Towards_Action_ECCV_2018_paper.pdf
> > [3] https://arxiv.org/abs/1904.07911
> > [4] https://arxiv.org/abs/1712.00377
> > [5] https://arxiv.org/abs/1902.01007
> > [6] https://arxiv.org/abs/1907.07355
> > [7] https://arxiv.org/abs/1908.10763
> > [8] https://arxiv.org/abs/1909.03683
> > [9] https://openreview.net/pdf?id=Bygh9j09KX
> > [10] https://arxiv.org/abs/1811.11155
> > [11] https://arxiv.org/abs/1903.06946
> > [12] https://arxiv.org/abs/1812.02725

---

### Official Review · AnonReviewer2 · 2019-10-24
**Official Blind Review #2**

**Rating:** 6

**Review:**

The paper describes a methodology for reducing model dependance on bias by specifying a model family of biases (i.e. conv nets with only 1x1 convs to model color biases), and then forcing independence between feature representations of the bias model and the a full model (i.e. conv nets with 3x3 convs to also model edges).

Overall the method is very interesting. A few very related recent works were missing: https://arxiv.org/pdf/1908.10763.pdf and https://arxiv.org/abs/1909.03683. These works provide different formulations for factoring out know "bias oriented" models (and focus more on NLP, although the second is applied to VQA). Although, I do appreciate that the bias models used in this work are slightly more general in terms of family than those studied before, they do encode significant intuition about the target bias to be removed and perhaps in this context, the methods cited above could also be compared. That being said, I don't feel the paper needs to provide this comparison given how recent those works are, but it would improve the quality of the paper.

One aspect that worries me about this paper is that most of the biases studied are synthetic, so specification of the bias family is trivial, in contrast to the works I mentioned above (where the bias model is potentially somewhat misspecified and needed to be discovered by different researchers). But I do really like the experiments on Imagenet-a.

Overall the paper presents an interesting contribution that would be useful for future study in reducing bias dependance in ml.



**Experience Assessment:**

I have published in this field for several years.

**Review Assessment: Checking Correctness Of Derivations And Theory:**

I assessed the sensibility of the derivations and theory.

**Review Assessment: Checking Correctness Of Experiments:**

I carefully checked the experiments.

**Review Assessment: Thoroughness In Paper Reading:**

I read the paper thoroughly.

---

> ### Author Response · Authors · 2019-11-15
> **Response to Reviewer2**
>
> We thank all the reviewers for their recognition of the task “interesting and important” (R1), and finding the proposed method “sound and interesting” (R2, R3). In particular, R3 has commented that “the paper does a really good job of defining the problem setting”. Yet, we find some lack of detail and clarity in our paper could potentially have led to misunderstanding our work -- we have clarified them in reviewer-specific responses and have updated the paper accordingly.
>
> We summarize our revisions to the paper as follows:
>
> Section 2:
> Based on the helpful comments of R1 and R3, we have clarified that
> * G may capture important signals as well as biases (Section 2.3, paragraph 3).
> * G can be easily defined in a broad class of problems in machine learning applications  (Section 2.3, paragraph 4).
>
> * We have updated additional related works provided by R2 (Section 2.3, paragraph 4).
>
> Section 3:
> * We have clarified that REBI’S does not suppress the signal captured by G due to its formulation as conditional independence (Section 3.1, Independence versus separation).
>
> Section 4:
> * As requested by R3, we have added additional comparison on vanilla BagNets and Stylized ImageNet augmentation [1] on ImageNet (Table 3).
> * There was a minor error in our evaluation code on ImageNet-A, and we have updated the correct values for every method (Table 3).
>
> * We have fixed typos throughout the paper.
>
> Our response to Reviewer 2 is as follows:
>
> #1 Most of the biases studied are synthetic and trivial.
>
> > Studying biases in models is difficult because it requires a dataset where bias is well-controlled (Section 4.1). We approach the difficulty by presenting two sets of experiments: Biased MNIST (synthetic but fully controlled biases) and ImageNet (realistic but less well-controlled biases). While the former may seem trivial, it allows us to prove that the novel algorithm REBI’S does work as intended on the simplest type of biases. With full control over the degree of bias, we can measure its performance on perfectly unbiased test data, which is infeasible to achieve in real-world datasets (i.e., all datasets are biased in their own manner [1]). The synthetic bias allows us to explicitly evaluate the bias-specific (color and texture) performances (Table 1). On ImageNet, we evaluate our method against realistic texture biases [2]. Experiments show that REBI’S does remedy biases towards texture and improves the generalization to the unbiased set and the ImageNet-A dataset (Table 3).
>
> #2 A few very related recent works were missing.
>
> > Thank you for mentioning the recent related works. We have included them in the revised paper. We believe extending our framework to the entailment task is an interesting future work.
>
> <References>
> [1] http://citeseerx.ist.psu.edu/viewdoc/download?doi=10.1.1.208.2314&rep=rep1&type=pdf
> [2] https://openreview.net/pdf?id=Bygh9j09KX

---

### Official Review · AnonReviewer1 · 2019-11-01
**Official Blind Review #1**

**Rating:** 6

**Review:**

This manuscript discusses the problem of bias shortcut employed by many machine learning algorithms (due to dataset problems or underlying effects of any algorithmic bias within an application). The authors argue that models tend to underutilize their capacities to extract non-bias signals when bias shortcuts provide enough cues for recognition. This is an interesting and important aspect of machine learning models neglected by many recent developments.

The only problem is that the paper seems to be a bit immature as the exemplar application is too naive for illustrating the idea. The authors’ idea is to assume that
‘there is a family of feature extractors, such that features learned by any of these extractors would correspond to pure bias. Then in order to learn unbiased features, the goal is to construct a feature extractor that is ''as different as this family" as possible’
in practice, their claim is texture and color are biases; one should learn shape instead of texture and color. So the family of feature extractors are the ones with small receptive field that can only capture texture and color. Therefore, what they eventually achieved is the unbiased feature extractor only learns the shape of object and avoids learning any texture and color.

So, the problem is, in practice, it is very hard to define the family of biased feature extractors. It really depends on the dataset and the goal. Texture and color, in general, are still important cues for object recognition, removing this information is NOT equivalent to removing bias. Just as a suggestion, the background scene might be a better definition of bias. However, with the proposal in this paper, it would be unclear how to define the family of feature extractors for describing background. Therefore, the solution given for this important problem seems to be too ad-hoc and not generalizable.

The second example (that does not have an experiments on) is action recognition; the family of biased feature extractors is 2D-frame-wise CNNs (object recognition). The authors claim that objects are biases for action recognition systems, but again a large part of action recognition is indeed object recognition. Many actions are defined based interaction of humans with objects (e.g., opening bottle or pouring water from bottle). Some objects may be instroducing bias in the task, but not all. Again, the proposed solution in this paper cannot disentangle this.

The authors need to survey previous texture-shape disentanglement works and then compare with those methods.




**Experience Assessment:**

I have published one or two papers in this area.

**Review Assessment: Checking Correctness Of Derivations And Theory:**

I assessed the sensibility of the derivations and theory.

**Review Assessment: Checking Correctness Of Experiments:**

I carefully checked the experiments.

**Review Assessment: Thoroughness In Paper Reading:**

I read the paper thoroughly.

---

> ### Author Response · Authors · 2019-11-15
> **Response to Reviewer1 (1/2)**
>
> We thank all the reviewers for their recognition of the task “interesting and important” (R1), and finding the proposed method “sound and interesting” (R2, R3). In particular, R3 has commented that “the paper does a really good job of defining the problem setting”. Yet, we find some lack of detail and clarity in our paper could potentially have led to misunderstanding our work -- we have clarified them in reviewer-specific responses and have updated the paper accordingly.
>
> We summarize our revisions to the paper as follows:
>
> Section 2:
> Based on the helpful comments of R1 and R3, we have clarified that
> * G may capture important signals as well as biases (Section 2.3, paragraph 3).
> * G can be easily defined in a broad class of problems in machine learning applications  (Section 2.3, paragraph 4).
>
> * We have updated additional related works provided by R2 (Section 2.3, paragraph 4).
>
> Section 3:
> * We have clarified that REBI’S does not suppress the signal captured by G due to its formulation as conditional independence (Section 3.1, Independence versus separation).
>
> Section 4:
> * As requested by R3, we have added additional comparison on vanilla BagNets and Stylized ImageNet augmentation [1] on ImageNet (Table 3).
> * There was a minor error in our evaluation code on ImageNet-A, and we have updated the correct values for every method (Table 3).
>
> * We have fixed typos throughout the paper.
>
> Our response to Reviewer 1 is as follows:
>
> #1 The authors assume that “there is a family of feature extractors (G), such that features learned by any of these extractors would correspond to pure bias.” In practice, it is very hard to define the family of biased feature extractors.
>
> > In Section 2.3, we have defined G as the set of “bias extractors except for ones that can also recognize [essential cues]”. In practice, we agree with R1 that defining G that precisely captures bias is difficult, and it is likely in many applications that any definition of G still encodes “important cues” along with biases (imperfect precision).
>
> Nonetheless, we argue that G does not need to precisely capture the bias. Our framework does not remove cues captured by G if they are essential for the target task. Note that REBI’S is built upon the “conditional independence” criterion (Section 3.1, “Independence versus separation” paragraph) that f(x) is encouraged to be independent of g(x), **given** the target task Y. Unlike independence that requires f(x) to ignore g(x) altogether, conditional independence allows f(x) to still retain cues captured by g(x) if they highly correlate with Y. In short, our goal is to **discourage over-reliance** towards cues captured by g(x) and push the model to learn other features that are useful for prediction. In the revision, we have improved the description of the precision and recall conditions for G in Section 2.3: those conditions are not meant as hard requirements but as desiderata. Even if they are not precisely met, conditional independence will do the rest of the job. Please also see our response to R3 because many of our points are relatable.
>
> Since G does not have to be perfect, it is easy to define G’s in many interesting application scenarios using prior knowledge on the relevant biases. We supply many examples below (also included in the response to R3). Action recognition models have been reported to rely heavily on static cues without learning temporal cues [2,3]; though the actual bias may not precisely be _any static cue_, we can still regularize the 3D convolutional networks towards better generalization across static cue biases by defining G to be the set of 2D convolutional architectures. It has been argued that visual question answering (VQA) models, too, rely overly on language biases rather than the visual cues (e.g. without looking at the image, one knows the answer to “what color is the banana” is “yellow”) [4]. We can define G as the set of models looking at the language modality only. Entailment models are biased towards the presence of certain words (e.g. when there are many “not”s, the sentence is “contradictory”), rather than really understanding the underlying meaning of sentences [5,6]. In this case, we can design G to be the set of bag-of-words classifiers [7,8]. In the revised paper, we have supplemented the above examples in Section 2.3.

---

> > ### Author Response · Authors · 2019-11-15
> > **Response to Reviewer1 (2/2)**
> >
> >
> > #2 In practice, their claim is texture and color are biases, yet they are still important cues for object recognition.
> >
> > > We do not claim texture and color to be biases in _all scenarios_. In Section 2.1, we have defined bias as “cues not essential for the recognition but correlated with the target Y” and the key property for a bias is that “intervening on [the bias] should not change [the target Y]”. In the biased MNIST experiments, we refer to color and texture as biases (B) because changing them do not change the semantics of the digits (Y). In the ImageNet experiments, we use the prior knowledge that image classifiers tend to be biased towards “local patterns” such as color and texture [9], though they may still be important cues for recognition. Essentially, our goal is to **discourage over-reliance** towards those cues rather than removing them altogether. As a result, we show improved performances in the realistic image classification task (Table 3).
> >
> > #3 Compare with previous texture-shape disentanglement works.
> >
> > > There are several works that attempt to disentangle texture and shape for the image generation task [10-12]. However, as R1 would agree, they are irrelevant because texture is not always a bias. REBI'S is conceptually better because its conditional independence does not remove all texture cues. As a similar line of work, Stylized ImageNet [9] attempts to achieve the same effect by augmenting texturized versions of the images during training. The aim is to make the model rely more on shape cues than texture. We have empirically compared against Stylized ImageNet (also requested by R3).
> >
> > Models                                                   	Biased           	Unbiased       IN-A
> > ResNet18                                           	        93.3                	85.8                	30.5
> > StylizedImageNet ResNet18                 	92.5                	87.6                	29.7
> > REBI’S (ResNet18-BagNet18)	        	93.7                	88.4                	31.7
> >
> > StylizedImageNet-trained ResNet18 shows improvements in reducing texture bias (improved “unbiased” accuracy from 85.8 to 87.6), but it does not increase the generalizability to the challenging natural adversarial examples (ImageNet-A accuracy from 30.5 to 29.7). REBI’S improves upon the StylizedImageNet-trained model both in terms of the unbiased and ImageNet-A accuracies (1.2 pp and 2.0 pp, respectively). We have updated the results in Table 3.
> >
> > <References>
> > [1] https://arxiv.org/abs/1610.02413
> > [2] http://openaccess.thecvf.com/content_ECCV_2018/papers/Yingwei_Li_RESOUND_Towards_Action_ECCV_2018_paper.pdf
> > [3] https://arxiv.org/abs/1904.07911
> > [4] https://arxiv.org/abs/1712.00377
> > [5] https://arxiv.org/abs/1902.01007
> > [6] https://arxiv.org/abs/1907.07355
> > [7] https://arxiv.org/abs/1908.10763
> > [8] https://arxiv.org/abs/1909.03683
> > [9] https://openreview.net/pdf?id=Bygh9j09KX
> > [10] https://arxiv.org/abs/1811.11155
> > [11] https://arxiv.org/abs/1903.06946
> > [12] https://arxiv.org/abs/1812.02725

---

### Comment · AnonReviewer3 · 2019-11-14
**Important concerns**

It seems that reviewer 1 and I have very similar concerns.

I would highly encourage the authors to address these while they can. I think it's crucial to answer these, and without a response, it's unlikely that I'll keep recommending an accept for the paper.

---

### Author Response · Authors · 2020-07-02
**To be presented at ICML 2020**

Dear Readers, Official Reviewers, and the Area Chair,

Thank you for your interest in this paper. Thanks to the constructive feedback from ICLR reviewers and the AC, we have further advanced the methodology and expanded the set of experiments over the past months. The new paper is accepted at ICML 2020 and will be presented in two weeks!

Paper (ICML version): https://arxiv.org/abs/1910.02806
Code: https://github.com/clovaai/rebias

Highlights of changes (ICLR->ICML):
- Simplification of ReBias: (1) remove unnecessary normalization (CKA) and (2) match the inner and outer objective for the minimax optimization.
- Experiments on action recognition (Kinetics with 3D CNNs).
- Comparison against further baselines that appeared after the ICLR 2020 deadline (RUBi and LearnedMixin).
- Code is released.

Thanks again for letting us improve our research, and hope you enjoy the new paper & code.

Thanks,
ReBias authors

---

### Decision · Program_Chairs · 2019-12-19

**Decision:**

Reject

**Comment:**

This paper provides and analyzes an interesting approach to "de-biasing" a predictor from its training set.  The work is valuable, however unfortunately just below the borderline for this year.  I urge the authors to continue their investigations, for instance further addressing the reviewer comments below (some of which are marked as coming after the end of the feedback period).